# Synthesis and Characterization of 1-Hydroxy-5-Methyltetrazole and Its Energetic Salts

**DOI:** 10.3390/molecules30132766

**Published:** 2025-06-27

**Authors:** Lukas J. Eberhardt, Maximilian Benz, Jörg Stierstorfer, Thomas M. Klapötke

**Affiliations:** Department of Chemistry, Ludwig-Maximilian University of Munich, 81377 Munich, Germany; lukebch@cup.uni-muenchen.de (L.J.E.); jstch@cup.uni-muenchen.de (J.S.)

**Keywords:** tetrazole, energetic materials, cycloaddition, oxidation

## Abstract

The objective of this work was the synthesis and characterization of novel, insensitive high explosives. 1-hydroxy-5-methyltetrazole served as both a scaffold and anion for preparing various nitrogen-rich energetic salts. The compounds were characterized using ^1^H and ^13^C NMR spectroscopy, high-resolution mass spectrometry, elemental analysis, low-temperature single-crystal X-ray diffraction, and IR spectroscopy. Thermal stability was investigated via differential thermal analysis (DTA). Sensitivities towards mechanical stimuli were measured using a BAM drop hammer for impact sensitivity and a BAM friction apparatus for friction sensitivity, employing one of six testing procedures. Energetic performance parameters were calculated using the EXPLO5 code, incorporating room-temperature X-ray densities and solid-state heats of formation obtained via CBS-4M calculations using the Gaussian 16 program.

## 1. Introduction

The majority of secondary explosives exhibit significant shortcomings. 1,3,5-Trinitrotoluene (TNT) is a melt-castable explosive, with a melting point of 80 °C. It is possible to melt TNT in a steam bath and cast it into its desired shapes; in doing so, an ideal filling is obtained. To be considered a melt-castable explosive, the melting temperature should be in the range of 75–115 °C, while decomposition temperatures should be higher than 180 °C [1,2]. TNT has been produced and used in explosive charges and formulations for over a century. Despite the affordability, accessibility, high decomposition temperatures, and low sensitivity towards mechanical stimuli, TNT suffers from two problems and therefore needs to be substituted. Firstly, TNT is toxic and a potential human carcinogen. It was shown that the groundwater and soil near production sites or training grounds are contaminated with TNT, by-products like dinitrotoluenes and degradation products, posing a threat to the environment [3,4,5]. The second problem is the weak detonation performance compared to other military explosives like RDX (1,3,5-trinitro-1,3,5-triazinane) or HMX (1,3,5,7-tetranitro-1,3,5,7-tetrazocane). RDX and HMX—while having a high detonation velocity—are toxic and pose an environmental threat on their own and through their degradation products [5,6]. Current research trends in energetic materials focus on developing more environmentally friendly explosives that are less toxic while maintaining or enhancing detonation performance and sensitivity. Another important consideration is the ‘green’ synthesis of new energetic materials, which should ideally have a high atom economy and use readily available starting materials. Furthermore, hazardous reagents and solvents should be avoided in the synthesis to ensure safer, more sustainable processes [7].

Due to their favorable energetic properties, tetrazoles have attracted considerable interest as scaffolds for high-energy-density materials (HEDMs). Among the azoles, tetrazoles exhibit the second highest heat of formation after pentazoles, resulting in good energetic performance. Their high nitrogen content ensures that the primary decomposition product is the non-toxic elemental dinitrogen [1,8]. 5-substituted tetrazoles are synthetically accessible via various protocols, including the 1,3-dipolar cycloaddition of nitriles with azides [9,10,11,12] or ring-closing reactions of azidoimines [13,14,15,16]. Introducing N-hydroxy functionalities into tetrazoles can significantly enhance the physicochemical properties of neutral hydroxytetrazoles and their salts compared to the parent compound. This is evidenced by an increase in oxygen balance and density and higher overall detonation velocities [17,18,19,20]. Hydroxytetrazoles can be synthesized primarily via two synthetic approaches. The direct oxidation of tetrazoles was first reported by Begtrup et al. in 1995; using sodium perborate in pivalic acid at 100 °C, 5H-tetrazole was converted to 1-hydroxytetrazole and 2-hydroxytetrazole in a 2:1 molar ratio and a combined yield of 50% [21]. In 1999, Giles et al. pioneered the use of Oxone^®^ (KHSO_5_·KHSO_4_·K_2_SO_4_), buffered at pH 7.5, for the direct oxidation of tetrazoles. In this case, ethyl tetrazole-5-carboxylate was selectively converted to ethyl-2-hydroxytetrazole-5-carboxylate [22]. The utilization of Oxone^®^ has now become standard methodology for the introduction of N-hydroxy functionalities in tetrazole rings. The oxidation of 5-substituted tetrazoles results in mixtures of regioisomers if the 5-substituent is an electron-donating group (EDG) [21]. However, if the 5-position bears an electron-withdrawing group (EWG), such as N_3_ [19], CO_2_Et [22], or NO_2_ [23], the reaction proceeds regioselectively towards N2. In addition to direct oxidation, 1-hydroxy-tetrazoles can be synthesized utilizing the azidooxime 1-hydroxytetrazole tautomerism. Azidooximes are readily available starting from aldehydes or nitriles. First, the aldehyde is converted to the corresponding oxime by reacting with hydroxylamine, followed by chlorination to form an oxime chloride. The cascade for nitriles is similar: the nitril reacts with hydroxylamine, forming an amidoxime, which can be diazotized with sodium nitrite in hydrochloric acid, yielding the oxime chloride. The subsequent reaction of the oxime chlorides with NaN_3_ results in the formation of the azidooxime, which is cyclized under acidic conditions, yielding 1-hydroxy-tetrazoles [20,24,25,26,27]. The synthesis of 1-hydroxy-5-methyl tetrazole was first documented by Bettinetti et al. in 1956, via the reaction of hydrazoic acid and ethyl nitrolic acid. In addition to the neutral compound, the silver salt was also prepared. However, neither compound was characterized completely, nor were their energetic parameters explored [28]. For safety reasons, in this study, 1-hydroxy-5-methyltetrazole was synthesized by the oxidation of 5-methyltetrazole.

## 2. Results and Discussion

Parts of this work were already published at the NTREM conference [29].

### 2.1. Synthesis

5-Methyltetrazole was first synthesized by Oberhummer in 1933 via the diazotization of acetimidohydrazide using ethyl nitrate [15]. The first reported synthesis of 5-methyltetrazole via 1,3-dipolar cycloaddition was carried out by Mihina et al. in 1950, achieving a yield of 76% from the reaction of acetonitrile with hydrazoic acid [30]. More recent methods use the reaction of sodium azide with acetonitrile in the presence of catalysts, such as NiFe_2_O_4_, Cu(II)-NaY zeolite, or CuO/aluminosilicate, in DMF, yielding up to 99% of the desired product [31,32,33].

In 2001, Sharpless et al. significantly improved the preparation of 5-substituted tetrazoles by performing the 1,3-dipolar cycloaddition of organic nitriles with sodium azide in water, using zinc Lewis acids as catalysts, a more environmentally friendly alternative to conventional protocols, which typically employ toxic organic solvents, such as DMF (Figure 1) [9]. As the aqueous solution is not acidic, minimal to no hazardous hydrazoic acid is released, unlike other protocols. The reaction is catalyzed by zinc Lewis acids such as ZnCl_2_, Zn(ClO_4_)_2_, and ZnBr_2_, the latter of which was found to give the highest yields. Due to the low reactivity of acetonitrile, an electron-rich nitrile, elevated temperatures (170 °C) are required, necessitating the use of a pressure tube or autoclave as the reaction vessel. Minor optimizations increased the yield from 75% to 90% by adjusting the stoichiometry from a 1:1.1 molar ratio of nitrile to azide to a 2:1 molar ratio, as well as keeping the amount of water during workup to a minimum. Given the high water solubility of 5-methyltetrazole, minimizing water content during the workup was crucial to enable efficient extraction. This was achieved using concentrated aqueous NaOH to precipitate Zn(OH)_2_ and concentrated aqueous HCl to protonate the sodium tetrazolate [9,12].

The subsequent oxidation of 5-methyltetrazole is carried out using a fourfold excess of Oxone^®^ in water at 40 °C (Figure 2). Oxone^®^ and tri-sodium phosphate dodecahydrate are added alternately over one hour, maintaining a pH of 7–8. No conversion was observed under unbuffered conditions. The maximum yield is achieved after three days. Increasing the reaction time or adding more equivalents of Oxone^®^ at the start or after three days did not increase the yield.

The reaction yields the regioisomers 1-hydroxy-5-methyltetrazole (**2**) and 2-hydroxy-5-methyltetrazole (Figure 2) in a 3:1 molar ratio, as determined by ^1^H NMR spectroscopy. Pure compound **2** is obtained by recrystallization using a mixture of acetone and *i*-hexane in a 52% yield.

A series of high-nitrogen salts (compounds **3**–**7**) were prepared by reacting compound **2** with the appropriate bases in water, ethanol, or a mixture of both solvents (Figure 3). For salts **4**, **5**, and **7**, equimolar amounts of base were used, whereas salt **3** was obtained using an excess of 2 M aqueous ammonia solution. The guanidinium salt **6** was synthesized by treating compound **2** with guanidinium carbonate. In the case of (D) and (E), heating the solution to 80 °C for 15 min was needed for conversion. Following solvent evaporation, all salts were obtained in good to quantitative yields. Detailed experimental procedures are found in the Appendix A.

### 2.2. X-Ray Diffraction

Single crystals suitable for X-ray diffraction were obtained by recrystallization from ethanol or water. Detailed information regarding the measurement and refinement of all compounds is provided in the Appendix A (Appendix A). The crystallographic data has been deposited in the Cambridge Structural Database (CSD) under CCDC deposition numbers 2,453,219 (for **2**), 2,453,217 (for **3**), 2,453,215 (for **4**), 2,453,218 (for **5**), 2,453,216 (for **6**), and 2,453,214 (for **7**).

Compound **2** crystallizes in the orthorhombic space group Pna2_1_ with a cell volume of 445.62(3) Å^3^ and four molecular units per cell. The cell constants are *a* = 9.6928(4) Å, *b* = 4.0112(2) Å, and *c* = 11.4614(5) Å. The recalculated density at 298 K is 1.465 g cm^−3^ (Figure 1A). Compared to 5-methyltetrazole (1.349 g cm^−3^ at 296 K), the introduction of the hydroxy group results in an increase of over 0.1 g cm^−3^ [34]. The molecule is almost planar, and the N-hydroxy function bends slightly out of plane, O1–N1–C1–N4 175.76°. The bond lengths in the aromatic ring (N1–N2 1.343 Å, N2–N3 1.286 Å, and N4–C1 1.318 Å), as well as the N-bonded hydroxide (O1–N1 1.356 Å), fall within the standard range. The molecules form strong hydrogen bonds, with an H1–N4 distance of 1.63(5) Å, leading to the formation of zig-zag chains. These chains are arranged alternately in the *a*–*c* plane, forming layers that stack along the *b* axis (Figure 1B).

Compound **3** crystallizes in the orthorhombic space group Pbca with a cell volume of 2131.1(2) Å^3^ and sixteen molecular units per cell. The cell constants are *a* = 13.1916(8) Å, *b* = 7.5481(4) Å, and *c* = 21.4023(10) Å, while the density is 1.460 g cm^−3^ at 298 K (Figure 2A). Compared to ammonium 5-methyltetrazolate (1.259 g cm^3^ at 298 K), the density is significantly higher [35]. The bend of the N-hydroxy function of the neutral compound **2** is lost, resulting in a planar structure, with the O1–N1–N2–N3 torsion angle measuring 179.97°. In all the salts **3**–**7**, the N–O bond is shortened by approximately 2 pm compared to the neutral compound. Each ammonium cation is surrounded by four or five hydroxytetrazolate anions, forming three hydrogen bonds to neighboring oxygen atoms in the range of O1–H5C 1.75 Å to O2–H5A 1.98 Å (Figure 2B).

Compound **4** crystallizes in the triclinic space group P-1 with a cell volume of 594.11(8) Å^3^ and four molecular units per cell. The cell constants are *a* = 7.3821(5) Å, *b* = 8.2511(6) Å, and *c* = 10.2099 Å, while the density is 1.461 g cm^−3^ at 298 K (Figure 3A). The molecules form chains in the b-c plane, with hydroxylammonium cations bridging two 5-methylhydroxytetrazolate anions via hydrogen bonds (O1–H2 1.682 Å; N4–H5C 1.968 Å). The two chains are arranged at a 180° rotation, with the methyl groups facing outwards, as they do not interact as much as the hydrogen bonds (N8–H5A 2.128 Å). The layers stack closely along the *b* axis due to strong hydrogen bonding (O1–H5B 2.128 Å) (Figure 3B).

Compound **5** crystallizes in the orthorhombic space group Pbca with a cell volume of 2436.0(7) Å^3^ and sixteen molecular units per cell. The cell constants are *a* = 14.632(2) Å, *b* = 6.9287(11) Å, and *c* = 24.028(5) Å, while the density is 1.404 g cm^−3^ at 298 K (Figure 4).

Compound **6** crystallizes in the monoclinic space group C2/c with a cell volume of 1516.87(16) Å^3^ and eight molecular units per cell. The cell constants are *a* = 11.9689(8) Å, *b* = 11.2025(7) Å, and *c* = 11.6752(6) Å, while the density is 1.367 g cm^−3^ at 298 K, the lowest density of all the herein investigated compounds (Figure 5A). Compared to guanidinium 5-methyltetrazolate (1.276 g cm^−3^ at 298 K), the introduction of the hydroxy group results in an increase of around 0.1 g cm^−3^ [36]. Each of the 5-methylhydroxytetrazolate anions is surrounded by four guanidinium cations, forming a network of medium to strong hydrogen bonds (O1–H5A 1.932 Å, N2–H5B 2.349 Å) (Figure 5B).

Compound **7** crystallizes with the inclusion of one water molecule in the triclinic space group P-1 with a cell volume of 569.36(11) Å^3^ and two molecular units per cell. The cell constants are *a* = 6.4887(7) Å, *b* = 6.9104(8) Å, and *c* = 13.2806(14) Å, while the density is 1.547 g cm^−3^ at 298 K, the highest density of all investigated compounds (Figure 6A). In the cation, the presence of different hybridized amino groups is evident. The C-bonded amines (C–NH_2_) are sp^2^ hybridized, as evidenced by the trigonal planar orientation of the protons (H12A–N12–H12B 118.4°) and the shorter C–N bond lengths (N11–C4 1.327 Å and N12–C5 1.325 Å). This can be attributed to the donation of electron density by the free-electron pair into the triazole rings. The N-bonded amine (N–NH_2_) is sp^3^ hybridized, with the free-electron pair located at the nitrogen N10. The hydrogen atoms form an almost ideal tetrahedral angle (H10A–N10–H10B 109.6°) with a longer bond length (N7–N10 1.405 Å). The molecules form chains where HTATOT cations bridge two 5-methylhydroxytetrazolate anions through hydrogen bonds (N4–H11B 2.11 Å, O1–H5 1.69 Å, and N2–H12A 2.20 Å). Two chains are arranged in an antiparallel fashion, rotated by 180°, with the methyl- and sp^3^-hybridized NH_2_ groups oriented outwards. The layers are staggered with the water molecules in between connecting the layers by hydrogen bonding (Figure 6B).

### 2.3. Physicochemical Properties

Impact sensitivities (ISs) and friction sensitivities (FSs) were determined according to BAM (Bundesanstalt für Materialforschung und-prüfung) standards, using a BAM drop hammer and friction apparatus, applying the 1 of 6 method [37,38]. Table 1 lists the experimental values. Except for the neutral compound **2** with 3 J and 160 N, all other investigated compounds, **3**–**6**, are insensitive towards impact (>40 J) and show little sensitivity towards friction with hydroxylammonium salt **4** being the most sensitive (288 N), while the others are insensitive with 360 N or >360 N.

The thermal stabilities were determined using differential thermal analysis (DTA) at a heating rate of 5 °C min^−1^. Thermal stabilities were determined by differential thermal analysis (DTA) with a heating rate of 5 °C min^−1^. The guanidinium salt **6** has the highest decomposition temperature of 256 °C. 1-Hydroxy-5-methyltetrazole melts at 146 °C and begins to decompose at 194 °C, which is 60 °C lower than 5-methyltetrazole **1**, which decomposes at 254 °C [39]. Compound **4** has a melting point of 139 °C, after which it decomposes. The ammonium **3** and the hydrazinium salt **5** decompose at similar temperatures of 229 °C and 224 °C, respectively.

The EXPLO5 V7.01.01 code was used to calculate the energetic performance parameters, using X-ray densities recalculated to room temperature and solid-state heats of formation obtained via CBS-4M calculations using the Gaussian 16 program, Revisions A.04. The energetic performance improved, and the oxygen balance increased from −76% to −48% by oxidizing 5-methyltetrazole (**1**). The detonation velocity increased from 6683 m s^−1^ to 7343 m s^−1^ [34]. Further increases in the energetic parameters were achieved through salification, with compounds **3**, **4**, and **5** improving substantially to around 8000 m s^−1^, with a detonation pressure of between 212 and 230 kbar. Only compound **6** deteriorated, with a V_det_ of 7090 m s^−1^, though still outperforming TNT. 1-hydroxy-5-methyltetrazole has the highest density of 1.465 g cm^−3^, while the salts range from 1.367 to 1.461 g cm^−3^.

### 2.4. Toxicity

A comparative in silico toxicological assessment was conducted using the ProTox 3.0 platform to evaluate the acute toxicity (LD_50_), mutagenicity, carcinogenicity, and acute ecotoxicity (on fish, daphnids and algae) of compounds **2**–**6**, TNT (2,4,6-trinitrotoluene), RDX (1,3,5-trinitro-1,3,5-triazinane), and HMX (1,3,5,7-tetranitro-1,3,5,7-tetrazocane) [40,41,42]. The prediction accuracy for compounds **2**–**6** was comparatively low (23%) relative to TNT (69%) and RDX/HMX (68%), indicating the need for further experimental validation. Predicted LD_50_ values suggest that compounds **2**–**5** (890 mg kg^−1^) and especially compound **6** (3150 mg kg^−1^) exhibit significantly lower acute toxicity compared to TNT (607 mg kg^−1^), RDX (100 mg kg^−1^), and HMX (186 mg kg^−1^). All new compounds were predicted to exhibit non-acute ecotoxicity with 55–71% probability, in contrast to the commercial explosives, which were predicted to show acute ecotoxicity with similar probabilities (TNT 77%, RDX 50%, HMX 53%). Regarding genotoxicity, compounds **2**–**5** were predicted to be non-mutagenic (52–62% probability), while guanidinium salt **6** was predicted to be mutagenic with a 61% probability. By contrast, TNT, RDX, and HMX were all predicted to be mutagenic with a high probability (99%, 89%, and 87%, respectively). Predictions of carcinogenicity indicate that compounds **2**–**6** are active (with a probability of 58–69%), as are RDX and HMX (with probabilities of 86% and 87%, respectively). In contrast, TNT was predicted to be inactive (55%).

Overall, the newly synthesized compounds **2**–**6** demonstrate favorable properties in terms of predicted acute toxicity, ecotoxicity, and mutagenicity when compared to conventional explosives TNT, RDX, and HMX. However, they are worse in terms of predicted carcinogenicity compared to TNT. The full reports are given in the Appendix A.

## 3. Conclusions

1-Hydroxy-5-methyltetrazole was synthesized in two steps using common chemicals and water as the reaction solvent. The [2 + 3] cycloaddition of acetonitrile with sodium azide catalyzed by zinc bromide was optimized. The subsequent oxidation of 5-methyltetrazole with Oxone^®^ and sodium phosphate dodecahydrate as a buffer produced compound **2**. Five nitrogen-rich salts, ammonium (**3**), hydroxylammonium (**4**), hydrazinium (**5**), guanidinium (**6**), and HTATOT (**7**) were synthesized by reacting the respective bases with compound **2** in water or ethanol. The energetic properties of the 5-methyltetrazole scaffold could be drastically improved by oxidation and salification. Although the hydroxylammonium salt **4** performs well, it is unsuitable due to its low decomposition temperature of 141 °C. Guanidinium salt **6** is the most thermally stable with a decomposition temperature of 256 °C; however, its low detonation velocity of 7090 m s^−1^ renders it unsuitable as a replacement candidate. A promising candidate for a cost-effective, insensitive explosive (40 J, >360 N) is ammonium salt **3**, which has a decomposition temperature of 229 °C and a detonation velocity of 7982 m s^−1^. The best-performing compound is hydrazinium salt **5**, with a detonation velocity of 8109 m s^−1^ and a detonation pressure of 219 kbar. It is insensitive towards friction and impact (40 J, >360 N) and has a high decomposition temperature of 224 °C.

## Data Availability

The data presented in this study can be found in the article and in the Appendix A.

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
