# Peer review of "Synthesis and Characterization of 1-Hydroxy-5-Methyltetrazole and Its Energetic Salts"

_molecules, 2025, doi:10.3390/molecules30132766_

Round 1

Reviewer 1 Report

Comments and Suggestions for Authors

An interesting paper on synthesis of a tetrazole.  These classes of compounds have potential interest for next generation formulations, but there is a need for significant additional application work to prove their use.

A few general points in the introduction need to be made.  TNT is not a 'so-called' melt cast explosive; it IS a melt cast explosive.  The phrase being sought is 'is described as a melt cast explosive'.  So called implies it is not such a species.  A great deal is made of the toxicity of TNT and both RDX and HMX.  The toxicity varies with solubility so specifics need to be stressed as the properties of the species are important.  However if you are going to stress that feature then some assessment of the probable toxicity is essential.  This is missing and is  glaring gap as it does not answer the question posed in the introduction.  Are these greener materials?  A short section is needed and estimation tools exist.

All energetic materials are reactive and thus also bioactive so addressing this in the same manner as impact and friction is required.

The syntheses are carefully performed and analysed so there are no comments on that.  The physiochemical testing is fine with TNT used as a benchmark.  Perhaps RDX and HMX could also be quoted but that is not essential.

Comments on the Quality of English Language

Mostly fine but some wrong choices made - see above about 'so-called'.  Needs some polish.

Reviewer 2 Report

Comments and Suggestions for Authors

The manuscript by Eberhardt et al. describes the two-step synthesis of 1-hydroxy-5-methyltetrazole (HMT-OH) and the preparation/characterization of five nitrogen-rich energetic salts. Most experimental procedures are carefully executed, and crystallographic and thermal data support the structure/property discussion well. The paper, therefore, merits publication after the authors address the following minor points.

1) Double-check empirical formulas: entry for salt four currently reads C₂H₇N₅O₂, yet the structure contains one extra oxygen (hydroxylamine cation).
2) There are two sequential “Figure 2” captions; renumber figures consistently throughout.
3) Provide a typical reaction scale (g of 1, volume of water) and isolation details for 2 and 2-hydroxy isomers, which are helpful for reproducibility.
4)         Briefly justify the use of CBS-4M for heats of formation (error margin vs. G4 or W1), and specify the phase correction applied (g → s) before EXPLO5 input.
5)     Minor language & typos e.g., “environmentally threat” → “environmental threat” (l. 36); “stereoselectively towards N2” → “regioselectively toward N2” (l. 64); “decompostition” → “decomposition” (Table 1 legend). Careful proofreading will catch similar slips.

Reviewer 3 Report

Comments and Suggestions for Authors

Comments

  1. Abstract, the first sentence, “derived from 1-hydroxy-5-methyltetrazole” or the title “1-hydroxy-5-methyltetrazole”?
  2. Remove the words, such as “fully”, etc.
  3. What is “one of six standardized testing procedures”? Is there standard?
  4. What about the results of the calculated energetic performance?
  5. Pay attention to the superscripts and subscripts through the whole text.
  6. “2:1 ratio”, is it mole or mass ratio? The same to the others.
  7. Regarding the supplemented Tables S1, S2 and Figures S1 and others, are they published previously? If they are important, it’s better to move them to the main document.
  8. There are several MD simulation results, it’s better to provide the simulation details.
  9. “impact (IS) and friction (FS) sensitivities” should be “impact sensitivities (IS) and friction sensitivities (FS)”.
  10. For the energetic performance calculation, the pressure, temperature parameters should be provided.

Reviewer 4 Report

Comments and Suggestions for Authors

The manuscript by Klapötke and his coworkers describes the “Synthesis and Characterization of 1-Hydroxy-5-Methyltetrazole and its Energetic Salts”. Overall, the authors have done a thorough job in the design, synthesis, characterization, and detonation properties of the new materials discussed in this manuscript. I recommend that this manuscript be accepted in its current format for publication in Molecules. 

The authors have thoroughly addressed the synthesis of the second highest nitrogen-containing tetrazole and its importance in the field of energetic materials. In particular, synthesizing the selected regioisomer, 1-hydroxy-5-methyltetrazole, and its nitrogen-rich salts is of great interest for environmentally safe energetic materials. All these new materials were fully characterized, and most of their structures were analyzed using single-crystal X-ray diffraction studies. Many of these new nitrogen-rich salts exhibit good thermal stability and excellent detonation properties compared to TNT explosives. I am sure this work will attract significant attention from readers in the field of high-energy materials.  

Reviewer 5 Report

Comments and Suggestions for Authors

Review for the manuscript ID:

molecules – 3696149

Title: Synthesis and Characterization of 1-Hydroxy-5-Methyltetrazole and its
Energetic Salts
Authors: Lukas Eberhardt, Maximilian Benz, Joerg Stierstorfer, Thomas M. M
Klapötke *
Applied Chemistry
https://www.mdpi.com/journal/molecules/sections/applied-chemistry
Synthesis, Characterization, and Application of Highly Energetic Materials
https://www.mdpi.com/journal/molecules/special_issues/TYZ6DR1007

The manuscript content is well suited for the specific focus and scope of current special issue: „Synthesis, Characterization, and Application of Highly Energetic Materials“.

In this manuscript authors reported the synthesis and characterization of novel insensitive high explosives derived from 1-hydroxy-5-methyltetrazole as anion and various nitrogen-rich energetic salts.

The detailed structure of prepared new salts were supported by modern spectroscopic methods and further determined via X-ray diffraction analysis. Energetic performance, thermal behavior, friction sensitivity and calculated detonation performance were provided for all synthesized salts. Experimental data analysis revealed that guanidinium salt was the most thermally stable (dec. temp. 256), but demonstrated a lower performance characteristics. The best energetic characteristics authors found for hydrazinium salt (detonation velocity 8109 m s-1 ), which was insensitive towards friction and impact and has a high decomposition temperature.

The current paper can be interesting and useful for the specialists in energetic materials area.

In general, the manuscript is rationally prepared, clearly written, technically well-re-checked, however, it still needs some minor corrections for the text improvement and references part re-constructing according to the journal requirements.

Manuscript Title and Abstract and Keywords: all three are good.

Figures and Tables are also good quality.

Reviewer can recommend just a few small corrections for the manuscript text improving

(Details are listed in attached PDF file)

After listed minor corrections will be done, the current manuscript can be accepted for publication.
